# A Survey on Massive MIMO Systems in Presence of Channel and Hardware Impairments

**DOI:** 10.3390/s19010164

**Published:** 2019-01-04

**Authors:** Zahra Mokhtari, Maryam Sabbaghian, Rui Dinis

**Affiliations:** 1School of Electrical and Computer Engineering, University of Tehran, Tehran 1439957131, Iran; z.mokhtari@ut.ac.ir; 2Instituto de Telecomunicações, 1049-001 Lisbon, Portugal; rdinis@fct.unl.pt

**Keywords:** massive MIMO, hardware impairments, doubly dispersive channels, channel aging

## Abstract

Massive multiple input multiple output (MIMO) technology is one of the promising technologies for fifth generation (5G) cellular communications. In this technology, each cell has a base station (BS) with a large number of antennas, allowing the simultaneous use of the same resources (e.g., frequency and/or time slots) by multiple users of a cell. Therefore, massive MIMO systems can bring very high spectral and power efficiencies. However, this technology faces some important issues that need to be addressed. One of these issues is the performance degradation due to hardware impairments, since low-cost RF chains need to be employed. Another issue is the channel estimation and channel aging effects, especially in fast mobility environments. In this paper we will perform a comprehensive study on these two issues considering two of the most promising candidate waveforms for massive MIMO systems: Orthogonal frequency division multiplexing (OFDM) and single-carrier frequency domain processing (SC-FDP). The studies and the results show that hardware impairments and inaccurate channel knowledge can degrade the performance of massive MIMO systems extensively. However, using suitable low complex estimation and compensation techniques and also selecting a suitable waveform can reduce these effects.

## 1. Introduction

Increasing number of users and strong demand for high data rate have forced researchers to increase the capacity and spectral efficiency of next generation wireless communications. The required data rate for 5th generation (5G) cellular communication is predicted to be 10 Gbps [1]. Different technologies for increasing the capacity of 5G, such as small cell communications, millimeter wave communications, and massive multiple input multiple output (MIMO) technology, have been suggested [2]. Small cell communication increases spectral efficieny by increasing the ability of frequency reuse, since it utilizes small cells such as femto cells and pico cells [3]. Millimeter wave communication, on the other hand, provides a large frequency band from 30 GHz to 300 GHz to increase the capacity of 5G [4]. In massive MIMO technology, a large number of antennas is used at the BS which increases the spatial resolution and consequently increases the spectral and power efficiency of the communications systems. Each of these technologies can separately increase the capacity of wireless communication systems considerably, and using them together can increase the capacity of the system even more [2]. However, these technologies face some challenges which have to be solved to reach the efficient implementation stage.

In this paper our focus is on massive MIMO systems. MIMO systems have attained considerable attention in the past two decades. Since they increase the capacity and reliability of the communication systems, they have been employed in many wireless standards [5]. Early studies on MIMO systems were on point-to-point MIMO where two systems with multiple antennas communicate with each other [6]. Later, attention was drawn to multi-user MIMO (MU-MIMO), where a BS with multiple antennas communicates with several single-or multiple-antenna mobile stations (MS). MU-MIMO has some advantages when compared with point-to-point MIMO. For example, expensive digital processors and power amplifiers are at the BS side and the MSs can have single antennas and be inexpensive [5,7]. However, MU-MIMO with nearly the same number of BS antennas and number of MS and frequency division duplex (FDD) mode transmission is not a scalable technology [7]. In other words, this technology is not responsive to the increasing number of users and the high data rate demand. Therefore, massive MIMO systems have been suggested. Massive MIMO was first suggested in 2010 in [8]. These systems are like MU-MIMO except that there is large number of antennas at the BS (100 antennas or more). In fact, in massive MIMO systems the number of BS antennas is much larger than the number of users. It is shown that in massive MIMO systems, as the number of BS antennas tends to infinity, the effect of small scale fading vanishes, the number of users in a cell becomes independent of the cell size, and the spectral efficieny will be independent of the bandwidth [5]. In fact massive MIMO relies on the law of large numbers to average out noise and small scale fading [7]. Regarding these advantages, massive MIMO has also been applied to wireless sensor networks (WSN), where the decision fusion center is equipped with large number of antennas [9,10,11,12,13]. The results show that massive MIMO can improve the performance of WSNs, and the power saving in these networks replicates the power savings obtained in massive MIMO mobile communications.

References [14,15] provide an overview on 5G technologies and signal processing techniques, while [16] gives a tutorial on modulation formats of 5G. There are also some tutorial papers specifically on massive MIMO systems. Reference [17] gives survey on channel measurement and modeling in massive MIMO systems. It also discusses some typical application scenarios along with key techniques in the physical and network layers. In [18,19], surveys on massive MIMO challenges from a detection and linear precoding perspective have been provided, respectively. Reference [20] gives a survey on pilot contamination and identifies possible sources of it. The authors of [21] provided an overview on some myths in massive MIMO systems and explained why they are not true. In this paper we provide a comprehensive overview on the effect of channel and hardware impairments in massive MIMO systems. First we introduce the basic concepts of massive MIMO systems and explain how they give the ability to the users to use the same physical layer resources (frequency band and/or time slots), simultaneously. After studying advantages and challenges of massive MIMO systems, we will give a brief review of the candidate waveforms for massive MIMO systems and introduce the two strong candidate waveforms: Orthogonal frequency division multiplexing (OFDM) and single-carrier frequency domain processing (SC-FDP). Next, we will specifically focus on the issue of hardware impairments and channel time variation in massive MIMO systems considering the two main waveform candidates. Since in massive MIMO systems we have large number of antennas at the BS, we have to use low-cost RF chains to have a cost effective system. This results in hardware impairments and degrades the performance of the system. Also, the performance of massive MIMO systems is very dependent on channel knowledge at the BS side. Due to Doppler effects, the channel varies in time (especially for high speed environments) and, therefore, estimating the channel and following its time variations is difficult. Therefore, paying attention to these two issues in massive MIMO is very important. Table 1 summarizes the challenges of massive MIMO systems studied in this paper and previous surveys.

The rest of the paper is organized as follows. In Section 2 the basic concepts of massive MIMO are explained. Section 3 explains the advantages and challenges of massive MIMO systems. Section 4 introduces the candidate waveforms for massive MIMO systems. Section 5 and Section 6 address the hardware impairment and channel variation issues, respectively. Finally, Section 7 concludes the paper.

### Notations

In the rest of the paper, we present the vectors and matrices in bold. We use ${\left(.\right)}^{*}$, ${\left(.\right)}^{H}$, and ${\left(.\right)}^{-1}$ to denote conjugate, conjugate transpose, and inverse of the argument, respectively. $I_{K}$ stands for the K×K identity matrix. We use ln(.), $\log_{2}\left(.\right)$, and $\lim_{x\to \infty }f\left(x\right)$ to show the natural logarithm function, logarithm function with base 2, and limit of function f(x) when its argument tends to infinity.

## 2. Basic Concepts

In typical massive MIMO systems, each cell has a BS with large number of antennas, which allows the simultaneous utilization of resources (i.e., frequency band and/or time slots) by different users in the cell. In the following we will introduce the system model and explain the basic concepts of massive MIMO systems in the uplink and the downlink. For simplicity, we consider a single cell scenario with flat fading channels. The extension to frequency selective channels will be straightforward when modulations like OFDM and SC-FDP are employed. Without loss of generality, we assume a BS with *M* antennas and *K* single atenna users.

### 2.1. Uplink Transmission

Figure 1 shows uplink transmission in massive MIMO systems. In the uplink, the users transmit their data and after passing through the channel, the received signals at the BS antennas are passed through a detection block to detect the data of each user. The M×1 vector of received signal at the BS is
(1)$$y^{\left(ul\right)}=\sqrt{p^{\left(ul\right)}}H^{\left(ul\right)}x^{\left(ul\right)}+J^{\left(ul\right)},$$
where $y^{\left(ul\right)}=\left[y_{1}^{\left(ul\right)},y_{2}^{\left(ul\right)},...,y_{M}^{\left(ul\right)}\right]$ and $H^{\left(ul\right)}$ is the M×K channel matrix between the BS antennas and the *K* users in the uplink. The *m*th row and *k*th column element of this matrix is the channel coefficient between the *m*th BS antenna and the *k*th user, and is defined as $h_{m,k}^{\left(ul\right)}=\sqrt{d_{k}^{\left(ul\right)}}g_{m,k}^{\left(ul\right)}$, where $d_{k}^{\left(ul\right)}$ and $g_{m,k}^{\left(ul\right)}$ indicate the large scale and small scale fadings, respectively. The parameters $J^{\left(ul\right)}$, $x^{\left(ul\right)}=\left[x_{1}^{\left(ul\right)},x_{2}^{\left(ul\right)},...,x_{K}^{\left(ul\right)}\right]$, and $p^{\left(ul\right)}$ are the M×1 noise vector, K×1 vector of the data sent by the *K* users, and the uplink transmission power of the users, respectively. In [22] it is shown that when M>>K, then the columns of the matrix $H^{\left(ul\right)}$ become orthogonal and therefore with simple linear detectors the multi-user interference can be removed. In other words if we rewrite $H^{\left(ul\right)}$ as $H^{\left(ul\right)}=G^{\left(ul\right)}{\left(D^{\left(ul\right)}\right)}^{\frac{1}{2}}$, where $G^{\left(ul\right)}$ is a M×K matrix with elements ${\left[G^{\left(ul\right)}\right]}_{m,k}=g_{m,k}^{\left(ul\right)}$ and $D^{\left(ul\right)}$ is a K×K diagonal matrix with diagonal elements $d_{k}^{\left(ul\right)}$, then with MF detector whose matrix is ${\left(H^{\left(ul\right)}\right)}^{H}$, we have
$$\frac{{\left(H^{\left(ul\right)}\right)}^{H}H^{\left(ul\right)}}{M}={\left(D^{\left(ul\right)}\right)}^{\frac{1}{2}}\frac{{\left(G^{\left(ul\right)}\right)}^{H}G^{\left(ul\right)}}{M}{\left(D^{\left(ul\right)}\right)}^{\frac{1}{2}}\approx D^{\left(ul\right)},\;M\gg K.$$

Since $D^{\left(ul\right)}$ is a diagonal matrix, it can be concluded that the interference between the users is completely removed. Therefore, it has been shown that the capacity of the *k*th user is [5]
(2)$$C_{k}^{\left(ul\right)}=\log_{2}\left(1+Mp^{\left(ul\right)}d_{k}^{\left(ul\right)}\right).$$

For a single-input single-output (SISO) system with similar conditions the capacity is
(3)$$C_{1}^{\left(ul\right)}=\log_{2}\left(1+p^{\left(ul\right)}d_{1}^{\left(ul\right)}\right).$$

Comparing (Equation 2) and (Equation 3), we can conclude that in massive MIMO systems the users can reduce their transmit power by *M* to have a performance similar to the SISO system. However, this is for the case where we have perfect channel knowledge and with imperfect channel knowledge this factor reduces [22].

We should note that other linear detectors like zero forcing (ZF) and minimum mean square error (MMSE) can also be used in massive MIMO systems. These detectors have similar behaviour with large number of BS antennas, being able to reduce substantially the multi-user interference. The matrix of ZF and MMSE detector, with respect to the uplink channel matrix, are defined as (Equation 4) and (Equation 5), respectively.
(4)$$W_{D}^{ZF}={\left({\left(H^{\left(ul\right)}\right)}^{H}H^{\left(ul\right)}\right)}^{-1}{\left(H^{\left(ul\right)}\right)}^{H}.$$
(5)$$W_{D}^{MMSE}={\left({\left(H^{\left(ul\right)}\right)}^{H}H^{\left(ul\right)}+\frac{\sigma_{\eta }^{2}}{\sigma_{x}^{2}}I_{k}\right)}^{-1}{\left(H^{\left(ul\right)}\right)}^{H}.$$

In (Equation 5), $\sigma_{\eta }^{2}$, $\sigma_{x}^{2}$, and $I_{k}$ are the variance of the noise, variance of the signal, and the K×K identity matrix.

### 2.2. Downlink Transmission

Figure 2 shows downlink transmission in massive MIMO systems. In the downlink, the data which have to be sent to each user are passed through a precoding block at the BS. Then the signal of each BS antenna is transmitted through the channel and received by the users. The precoder is employed to direct the signal of each user to its corresponding user. All the discussions for the uplink can be extended to the downlink with the only difference being the fact that in the downlink we have a precoding block instead of the detection block. As an example we assume to use MF precoder and therefore the M×1 vector of the precoded signal at the BS is
(6)$$y^{\left(dl\right)}={\left(H^{\left(dl\right)}\right)}^{H}x^{\left(dl\right)},$$
where $y^{\left(dl\right)}=\left[y_{1}^{\left(dl\right)},y_{2}^{\left(dl\right)},...,y_{M}^{\left(dl\right)}\right]$ and ${\left(H^{\left(dl\right)}\right)}^{H}$ and $x^{\left(dl\right)}=\left[x_{1}^{\left(dl\right)},x_{2}^{\left(dl\right)},...,x_{K}^{\left(dl\right)}\right]$ are the M×K MF precoder matrix and K×1 vector of users’ data, respectively. $H^{\left(dl\right)}$ is the K×M downlink channel matrix between the BS antennas and the *K* users. The *k*th row and *m*th column element of this matrix is the channel coefficient between the *m*th BS antenna and the *k*th user and is defined as $h_{k,m}^{\left(dl\right)}=\sqrt{d_{k}^{\left(dl\right)}}g_{k,m}^{\left(dl\right)}$, where $d_{k}^{\left(dl\right)}$ and $g_{k,m}^{\left(dl\right)}$ are the large scale and small scale fadings, respectively. In this case when M>>K, the rows of the matrix $H^{\left(dl\right)}$ become approximately orthogonal and, therefore, a simple linear precoder can mitigate multi-user interference. In other words, if we rewrite $H^{\left(dl\right)}$ as $H^{\left(dl\right)}={\left(D^{\left(dl\right)}\right)}^{\frac{1}{2}}G^{\left(dl\right)}$, where $G^{\left(dl\right)}$ is a K×M matrix with elements ${\left[G^{\left(dl\right)}\right]}_{k,m}=g_{k,m}^{\left(dl\right)}$ and $D^{\left(dl\right)}$ is a K×K diagonal matrix with diagonal elements $d_{k}^{\left(dl\right)}$, then after passing the precoded signal through the channel we have [5]
$$\frac{H^{\left(dl\right)}{\left(H^{\left(dl\right)}\right)}^{H}}{M}={\left(D^{\left(dl\right)}\right)}^{\frac{1}{2}}\frac{G^{\left(dl\right)}{\left(G^{\left(dl\right)}\right)}^{H}}{M}{\left(D^{\left(dl\right)}\right)}^{\frac{1}{2}}\approx D^{\left(dl\right)},\;M\gg K.$$

Since $D^{\left(dl\right)}$ is a diagonal matrix, the interference between the users is completely removed and each user can receive its desired signal.

We should note that other linear precoders such as ZF and MMSE are used in massive MIMO systems. These precoders have similar behaviour at large number of BS antennas and remove the interference between the users very well. The matrix of ZF and MMSE precoder, with respect to the downlink channel matrix, are defined as (Equation 7) and (Equation 8), respectively.
(7)$$W_{P}^{ZF}={\left(H^{\left(dl\right)}\right)}^{H}{\left(H^{\left(dl\right)}{\left(H^{\left(dl\right)}\right)}^{H}\right)}^{-1}.$$
(8)$$W_{P}^{MMSE}={\left(H^{\left(dl\right)}\right)}^{H}{\left(H^{\left(dl\right)}{\left(H^{\left(dl\right)}\right)}^{H}+\frac{\sigma_{\eta }^{2}}{\sigma_{x}^{2}}I_{k}\right)}^{-1}.$$

## 3. Advantages and Challenges

In this section we will introduce the advantages of this technology and explain its open problems and challenges which need further investigations.

### 3.1. Advantages

The advantages of massive MIMO are listed below [7]:*High spectral efficiency*: In massive MIMO, using large number of BS antennas and having aggressive spatial multiplexing gives the ability to the users to simultaneously use the same frequency band without interference. This increases the spectral efficiency of the system.*High power efficiency*: Since in massive MIMO we have large number of BS antennas, the energy can be concentrated in a small area and this increases the power efficiency. In these systems the users can reduce their transmit power as we increase the number of BS antennas, while being able to have a performance similar to the SISO case.*High degree of freedom*: Massive MIMO systems have high degree of freedom. For example if we have 200 antennas at the BS and 20 single antenna users, then we have 180 degree of freedom which can be used to help reduce the interference. In other words, we can design the transmitted signals from the BS antennas in a way that all these signals are added constructively at the users and destructively almost anywhere else.*Enabling reduction of latency*: Fading limits the performance of wireless communications systems and makes building low latency links hard. When a user experiences a fading dip it has to wait for suitable change of the channel before receiving the data. However, massive MIMO systems avoid fading dip by having large number of BS antennas and, therefore, enable reduction of latency.

### 3.2. Challenges

Massive MIMO technology as any other new technology, faces some problems and challenges. To make this technology reach the efficient implementation stage, these challenges need further study and investigation. In the following we will explain some of the main challenges in massive MIMO systems [5,7].
*Antenna array design*: Configuration and orientation of antenna arrays in massive MIMO systems is a problem which needs further studies. For example it should be investigated whether 2 dimensional or 3 dimensional antennas are suitable. Also having centralized or distributed antenna arrays and the distance between the antennas and the mutual coupling effect should be studied.*Implementation in FDD mode*: The performance of massive MIMO systems is very dependent on the channel knowledge at the BS. In the uplink the users send orthogonal pilots and the BS estimates the channel of each user considering the pilots. In the downlink of conventional MIMO systems, the BS sends pilots to the users and the users feedback the channel information to the BS, after estimating it. However, since the number of resources needed for channel estimation is proportional to the number of antennas at the transmitter, estimating the channel in downlink of massive MIMO systems where we have large number of antennas at the BS is not suitable and it requires many resources. Also in these systems, the number of channels which each user has to estimate is large and, therefore, a large amount of information has to be fed back to the BS. One solution is to use time division duplex (TDD) mode and take advantage of channel reciprocity to estimate the downlink channel in the uplink. However, this mode also has some drawbacks, for example the hardware impairments in the BS and user side may affect channel reciprocity, not to mention possible channel variations between the time it is estimated and the time it is used (i.e., channel aging effects). Also many current systems work with FDD mode, and for massive MIMO to be compatible with these systems we need to be able to implement it in FDD mode. So, implementing massive MIMO in FDD mode with low training and feedback overhead is an issue that needs further studies.*Pilot contamination*: In TDD mode, each user sends an orthogonal pilot for channel estimation. Since the coherence time of channel is limited, there is a limited number of orthogonal pilots and therefore users in other cells have to reuse these pilots. So, the estimated channel of a specific user will be contaminated with a linear combination of the channel of other users which use the same pilot. Since the beamforming is done considering these estimated channels, an interference between users who use the same pilot will occur.*Channel model*: Most of the works on massive MIMO systems are done with the assumption of time invariant flat rayleigh fading channels. Since in OFDM systems the channel over each sub-carrier is considered flat, the studies on flat fading channels can be extended to OFDM systems. However, in many practical scenarios, the channel is frequency selective and time variant. The channel coefficients can also have different distributions. So, studying these systems in more realistic environments is an issue which needs further investigations.*Waveform*: In 5G, to reach the required data rate and latency expected for future wireless communications the structure of the wireless communication needs significant changes. In other words, 5G is not a simple extension of 4G and it should use some new technologies and integrate different wireless access technologies [23]. Massive MIMO is one of these technologies. In each of these technologies the suitable waveform in different scenarios has to be investigated. Although in 2016 3GPP decided to study various features of 5G new radio (NR) assuming OFDM, but this can change if significant gains can be demonstrated by any other waveform [24,25,26]. Some of the reasons that OFDM is selected for NR uplink and downlink are [27]
-OFDM is a scalable waveform with low implementation complexity,-Compatibility with multi-antenna technologies,-High spectral efficiency.However, OFDM has some drawbacks such as lower frequency localization and high PAPR. Therefore, although an early decision was taken to use the OFDM as 5G waveform, the exact waveform has not yet been decided and other waveforms such as SC-FDMA (also denoted DFT spread OFDM) are also being investigated [24].*Hardware impairments*: Since in massive MIMO systems we have large numbers of antennas, we need to use low-cost hardware to have a cost effective system. This means that hardware impairments will appear, which affect the performance of the system. Therefore, analyzing the performance of massive MIMO systems in presence of hardware impairments and proposing low complexity techniques for compensating them is crucial.

## 4. Waveform Selection

In this section we will introduce some candidate waveforms for massive MIMO systems and explain some of the studies done in this field.

### 4.1. Candidate Waveforms

The candidate waveforms can be divided into two main categories, the multi-carrier waveforms and the single-carrier waveforms. Multi-carrier waveforms include OFDM and its multiple access form (OFDMA) and new versions, filter bank multi-carrier (FBMC) and generalized frequency division multiplexing (GFDM). Single carrier waveforms include traditional single-carrier (SC) and SC-FDP and its multiple access form (SC-FDMA). Since OFDM and SC-FDP and their multiple access forms are used in 4G and also they are the strong candidates for 5G, in the following we will introduce the system model of massive MIMO systems using these waveforms in details. More information on other waveforms such as FBMC, GFDM, and traditional SC can be found in [28,29,30].

#### 4.1.1. OFDM

Figure 3 shows the block diagram of massive MIMO system with OFDM waveform in both the uplink and the downlink. In the uplink, each user passes their data through an IFFT block and adds cyclic prefix (CP) to its data block and then sends it through the channel. CP is part of the end of the block which is added to the beginning of the block. At the BS, after removing the CP, the received signal of each BS antenna is passed through an FFT block and then the resulting signals are given to the detection block. This block separates the signal of each user and therefore each user’s signal is detected. In the downlink the data that has to be sent to each user is precoded in the BS and then the resulting signal of each BS antenna is passed through an IFFT block. After adding the CP, the signals are transmitted through the channel. At the receiver side, the CP is removed and each user passes their received signal through an FFT block and therefore detects its desired signal.

#### 4.1.2. OFDMA

OFDMA is the multiple access form of OFDM. In this waveform instead of allocating all the subcarriers to a user, the subcarriers are allocated with a specific pattern to the users. Figure 4 shows the block diagram of OFDMA massive MIMO system in the uplink and the downlink. If we assume *N* is the number of subcarriers allocated to a user and we have total of *K* users, then considering $\tilde{K}$ to be the number of users that use the same subcarriers, the total number of subcarriers will be $\tilde{N}=\frac{NK}{\tilde{K}}$. We should note that this is true if there are no guard subcarriers.

#### 4.1.3. SC-FDP

Block diagram of SC-FDP massive MIMO systems is shown in Figure 5. The block diagram of this system is similar to OFDM massive MIMO system except that the FFT and IFFT blocks are transferred from the user side to the BS. In fact, SC-FDP is a single-carrier waveform that processes the signal block wise. We should note that in conventional systems the block wise single-carrier waveform was called single-carrier frequency domain equalization (SC-FDE). However, in massive MIMO system since we have frequency domain equalization in the uplink and frequency domain precoding in the downlink of block wise single-carrier, we call this waveform single-carrier frequency domain processing (SC-FDP). We used the term processing to include both equalization and precoding.

#### 4.1.4. SC-FDMA

SC-FDMA is the multiple access form of SC-FDP. In this waveform the subcarriers are also allocated with a specific pattern to the users. Figure 6 shows the block diagram of SC-FDMA massive MIMO system in both uplink and downlink. In this case if we also assume *N* is the number of subcarriers allocated to a user and we have total of *K* users, where $\tilde{K}$ is the number of users that share the same subcarriers, then the total number of subcarriers will be $\tilde{N}=\frac{NK}{\tilde{K}}$. We should note that this is valid if we have no guard subcarriers in the system.

### 4.2. Studies on Multi-Carrier Waveforms

In this subsection we will briefly study some of the works done on massive MIMO systems with multi-carrier waveform.

In [22], uplink of OFDM massive MIMO systems has been considered. In this paper achievable rates of the system with MF, ZF, and MMSE detectors under perfect and imperfect channel state information have been derived. The results show that with perfect channel knowledge the users can reduce their transmit power by the number of BS antennas to have a rate similar to the SISO case. For imperfect channel knowledge case this reduction in power is by the square root of number of BS antennas. In [31], the power efficiency of OFDM massive MIMO system with power allocation and random distributed users has been computed in the uplink and downlink. In this paper a relation between optimum number of users and number of BS antennas and other system parameters was investigated. Also, to reduce the intercell and intracell interference a new detector and precoder called the full pilot zero forcing (P-ZF) was designed. In [32], the performance of OFDM massive MIMO system with MF and ZF precoder in the downlink was studied and lower bounds for the capacity were derived. The results indicate that at high SNRs, ZF precoder has better performance than MF precoder and at low SNRs it is the opposite. It is also concluded that ZF precoder is more suitable for high spectral efficiency and low power efficiency cases while MF precoder is more suitable for high power efficiency and low spectral efficiency cases. The authors of [33] studied the pilot contamination effect in OFDM massive MIMO downlink and proposed a precoder based on MMSE to reduce this interference. Also, some works addressed the problem of resource allocation in OFDM massive MIMO systems. For example, in [34] the authors have studied optimum power allocation and pilot assignment of users in uplink of OFDM massive MIMO system with MF precoder. Also, in [35] the optimum power allocation in the downlink to maximize the sum-rate was investigated. The results showed that as the number of BS antennas increases with respect to number of users, the gain achieved from power allocation strategies is decreased and equal power allocation will be the optimum.

### 4.3. Studies on Single Carrier Waveforms

In this subsection we will briefly study some of the works done on massive MIMO systems with single-carrier waveform.

In [36] BER of SC-FDP massive MIMO system in the uplink and in presence of frequency selective channel has been investigated. The results show that when the number of BS antennas is large, even with simple linear detectors, the BER performance of this system converges to the BER performance of a SISO system with flat fading channel. In [37], an iterative low complexity receiver for uplink of SC-FDP massive MIMO system has been designed. It was shown that even when the number of BS antennas is not very large, the performance of the designed receiver reaches that of single user system after only few iterations. The performance of different detectors in uplink of SC-FDP massive MIMO system has been studied in [38]. The authors of [39] studied the power efficiency of SC-FDP massive MIMO system with MF detector in the uplink. It was assumed that the users had multiple antennas and correlation between transmit antennas was considered. The results indicate that this system can have better performance than the system with single antenna users. In [40], the BER and PAPR of SC-FDMA and OFDMA in downlink of massive MIMO systems were studied through simulations. It was shown that SC-FDMA had lower PAPR and better BER performance than OFDMA.

## 5. Hardware Impairments

Since in massive MIMO systems we have large number of antennas at the BS, we should use inexpensive RF chains to have a cost effective system. This results in hardware impairments, which degrade the performance of the system. In the following we will introduce some of these hardware impairments.

### 5.1. Carrier Frequency Offset (CFO)

Using inexpensive local oscillators increases the probability of oscillator instability and causes carrier frequency offset, which are due to the mismatch between the oscillator of the transmitter and the receiver. The effect of CFO is modeled as a phase rotation in the received time domain signal or a frequency shift in the spectrum of the received frequency domain signal [41,42,43,44,45,46,47,48,49,50,51]. In massive MIMO systems, since we have multiple users we also have multiple CFOs to estimate and compensate. In centralized massive MIMO systems, since the BS antennas use a common oscillator, the number of CFOs that have to be estimated is equal to the number of users. However, in the distributed massive MIMO system each BS antenna has a separate oscillator which increases the number of CFOs to be estimated. So, the problem of estimating CFO is more severe in distributed massive MIMO systems [41].

Previous CFO estimation techniques which were proposed for conventional systems are not suitable for massive MIMO systems, since their complexity increases significantly with the number of BS antennas [42]. Therefore, studying the effect of CFO on the performance of massive MIMO systems and proposing low complexity algorithms for CFO estimation and compensation in these systems is vital.

In [41] the authors have investigated the effect of CFO on the sum-rate of centralized and distributed massive MIMO system in the uplink. In this paper, it is assumed that the channel is flat fading and perfect CSI is available. The results indicate that even with perfect CSI, if we do not have a good CFO estimation then the sum-rate of the system will degrade significantly, specially in the distributed case. The simulation based studies in [43] also confirm that in OFDM massive MIMO systems if the time and frequency offset are not estimated well, the performance of the system degrades significantly. In [42,44] the effect of residual CFO after CFO compensation on the sum-rate of massive MIMO systems in the uplink has been studied. In [42] the studies were done on flat fading channels with MF and ZF detectors and in [44] the studies were done on frequency selective channels with MF detector. It was concluded that with residual CFO, by increasing the number of antennas at the BS we can achieve an array gain similar to the no CFO case. In [45] the authors derived the sum-rate of OFDM and SC-FDP massive MIMO systems in the downlink and in presence of CFO. The sum-rates were analytically derived for MF and ZF precoders for both systems. The results show that OFDM massive MIMO system is more sensitive to CFO and its sum-rate is bounded even when the number of BS antennas tends to infinity while that of SC-FDP increases unlimitedly with the number of BS antennas. This is due to the fact that the interference caused by the CFO in the OFDM massive MIMO does not vanish as *M* increases.

In [46], the effect of channel knowledge at the user side on the sum-rate of SC-FDP massive MIMO system affected by CFO has been investigated. The sum-rates were derived for three different channel knowledge scenarios. In the first and the second scenario, the users have knowledge of the expectation of the total effective channel including the effect of CFO in time and frequency domain, respectively. In the third scenario the users only know the expectation of the effective channel without considering the effect of CFO. It was shown that in the first scenario, the effect of CFO can be completely compensated and the sum-rate is identical to the sum-rate of a fully synchronized system. It is also showed that the sum-rate in the second scenario is higher than the sum-rate in the third scenario when the number of BS antennas is finite. But, when the number of BS antennas tends to infinity the sum-rate in the second scenario is bounded and in the third scenario it tends to infinity. In summary, it can be concluded that CFO can have considerable effect on the performance of massive MIMO systems. Therefore proposing low complexity and accurate techniques for CFO estimation in these systems, specially with OFDM waveform is crucial.

In [47], a low complexity method for CFO estimation in SC-FDP massive MIMO systems was proposed. In this technique, each user sends first a pilot block to estimate its CFO at the BS and after CFO compensation another pilot block is sent for channel estimation. This technique has good performance and low complexity, which increases only linearly with the number of BS antennas. However, since a separate pilot block is used for the CFO estimation, the spectral efficiency decreases. To increase the spectral efficiency a blind CFO estimation technique was proposed [48]. In this technique null subcarriers are assigned to each user and it is shown that at large number of BS antennas the CFOs can be estimated blindly. However, the complexity of this technique is high being a cubic function of the number of BS antennas. A joint CFO and channel estimation technique for OFDM massive MIMO system was proposed in [49] to keep the spectral efficiency high. Here, the pilots sent for channel estimation are used for both CFO and channel estimation. Since the CFO estimation in this technique is done with linear grid search for each user, the complexity is high. In [50], a data aided technique for CFO estimation in OFDM massive MIMO system was proposed. In this paper it was assumed that the incident signal at BS from each user can be restricted within a narrow angular spread. In other words, when the number of BS antennas is large the users can be separated by their direction of arrival (DOA) and therefore the CFO of each user can be estimated. An iterative algorithm for estimating the CFO and DOA of each user was proposed in [50]. To have low complexity and keep the spectral efficiency high, an iterative channel and CFO estimation algorithm for SC-FDP and OFDM massive MIMO systems was proposed in [51]. In this technique, the same pilots are used for CFO and channel estimation to keep the spectral efficiency high. In this method, a least square channel estimation is done first, using the pilots. Next, the users are separated roughly using the estimated channel and then the CFO of each user is estimated individually as in a SISO system. Then, using the pilots and the estimated CFOs another channel estimation is done. With the new estimated channel, the users are separated again and another CFO estimation is done. This procedure is repeated until the MSE of channel and CFO converges to a desired value. Figure 7 and Figure 8 show the MSE of CFO estimation and MSE of channel estimation versus the number of iterations, respectively. The simulation results show that with only a few iterations the MSE of the channel estimation converges to that of the case where we have no CFOs. The results also show that this technique has better performance and lower complexity than [49]. The complexity of this technique increases linearly with the number of BS antennas.

### 5.2. Power Amplifier Distortion (PAD)

Power amplifier distortion appears when the input signal power is not in the linear range of the power amplifier [52]. There are different mathematical models used in the literature for power amplifiers such as the Rapp model, Saleh model, Ghorbani model, and some models with memory [53,54,55,56,57]. Power amplifiers with large linear range are expensive and therefore using inexpensive RF chains increases the PAD issue in massive MIMO systems.

In [45], the authors have derived the sum-rate of SC-FDP and OFDM massive MIMO systems in presence of PAD when the number of users is large. The PAPR and out-of-band radiation levels of these two systems have also been studied. The results show that when the number of users is small, PAPR and out-of-band radiation of SC-FDP is lower than OFDM. However, as the number of users increase the PAPR and out-of-bound radiation of SC-FDP increases and converges to that of OFDM. Also, the theoretical results show that the sum-rate of both systems for large number of users are identical and degrade significantly in compare to the case where we have no PAD. In fact, in the presence of PAD, the sum-rate of both systems saturate as the number of BS antennas increases. Therefore, handling the PAPR and PAD problem in massive MIMO systems is crucial.

Since working with constant envelope signals in these power amplifiers is easier, References [58,59] proposed precoders which produce signals with constant envelope in flat fading and frequency selective channels, respectively. In other words the precoder is designed in a way that the envelope of the precoded signal is constant, with the phase being used to reduce the interference levels. To improve the performance of the precoder proposed in [58] in terms of MUI reduction, the authors in [60] have proposed a precoder which produces signal with constant envelop where the signal envelope can be variable in a pre-defined period. The MUI reduction is better but the complexity and PAPR is larger than [58]. The precoders in [58,60] use sequential gradient descent algorithm to find the precoder coefficients, which has high complexity. To solve this issue, in [61] the authors proposed another technique which uses an approximate message passing algorithm. Results show that the PAPR of this technique is lower than [60] and higher than [58]. In [62], the authors have used the degrees of freedom of the system to reduce the PAPR. In this technique, the precoding, modulation and reducing the PAPR operations are done jointly. In summary, the techniques introduced in [58,59,60,61,62] are complex, since the precoder coefficients are dependent to the signal and, therefore, for every block of data the optimization problem for obtaining the precoder coefficients has to be solved. A simple technique for reducing the PAPR was proposed in [63] using a clipping operation. Since the clipping is a nonlinear operation, it causes distortion. Therefore, to overcome this problem, the BS antennas are divided into two groups. The first group is used for transmitting the clipped signal and the second group is used to send correction signals to compensate the distortion caused by clipping.

### 5.3. I/Q Imbalance

One of the hardware impairments in digital communication systems is in-phase/quadrature (I/Q) imbalance. In communication systems, the real part of the baseband signal is mixed with the high-frequency carrier and the imaginary part is mixed with the $\frac{\pi }{2}$ phase-shifted version of the carrier (i.e., a quadrature carrier). In an ideal system, the amplitude gain of the mixers for real and imaginary parts are identical and the difference between the in-phase and quadrature carriers is exactly $\frac{\pi }{2}$. However, in practical systems, mismatches between the amplitude and phase of the in-phase and quadrature components occur, which causes I/Q imbalance [64]. I/Q imbalance can occur at the transmitter and the receiver and is usually modeled as linear combination of the signal and its complex conjugate [65]. For example if *x* is the signal to be transmitted then the signal effected by the transmitter I/Q imbalance is modeled as
(9)$$y=x+\alpha x^{*},$$
where $x^{*}$ is complex conjugate of *x* and α is a real number smaller than 1.

For massive MIMO systems with I/Q imbalance, the CSI knowledge is not sufficient and I/Q imbalance needs to be estimated and compensated. In [66] a new precoder which jointly compensates I/Q imbalance and multi-user interference was proposed, which processes the real and imaginary part of the signal, separately. The results show that in massive MIMO systems with regularized ZF (RZF) precoder, the effect of I/Q imbalance does not vanish even when the number of BS antennas tends to infinity. However, with the new proposed precoder, this effect is vanished as the number of BS antennas increases. In [67], the effect of I/Q imbalance on channel estimation and channel reciprocity of massive MIMO systems in TDD mode were studied. Therefore, channel estimation in the uplink and achievable rate in the downlink were investigated in presence of both BS and user I/Q imbalance. The results show that only the receiver I/Q imbalance at the BS effects the channel estimation accuracy. It is also proved that the downlink sum-rate is limited either by the transmit and receive I/Q imbalance at the BS or by the receive I/Q imbalance at the user side.

### 5.4. Phase Noise

Phase noise is the most important parameter in many oscillators. In an ideal oscillator, the spectrum is line at the carrier frequency, but in practical systems the spectrum is non zero at the vicinity of the carrier frequency [68]. Therefore, in the ideal case the spectrum of the desired signal is convolved with Dirac function, and it is downconverted or upconverted without any change. But, in practical systems the spectrum of the signal at the output of the oscillator changes and, since the spectrum is expanded, an interference between the desired signal and the signal in adjacent channel will occur [68]. Therefore, analyzing the effect of phase noise and compensating it in massive MIMO systems is vital. Several models for phase noise have been proposed in the literature such as Leeson’s model, linear time invariant model, and linear time variant model [69,70,71].

In [72], the effect of phase noise in uplink of single-carrier massive MIMO system with imperfect channel knowledge has been investigated. The results show that this noise causes channel aging effects and, therefore, the time duration for data transmission and the number of supported users has to be reduced. However, this noise does not effect the array gain. In [73], the effect of phase noise on downlink sum-rate of OFDM massive MIMO system with MF and ZF precoder was investigated. It is shown that, although MF precoder has lower sum-rate per user, it is more robust to phase noise than ZF precoder. This is also true for the case where we have channel estimation errors. In [74], the authors investigated the effect of phase noise in single input multiple output (SIMO) and multiple input single output (MISO) systems. Two configurations were considered for these systems. In the first configuration all the RF chains use a common local oscillator and in the second configuration each RF chain has a separate local oscillator. The second-order term in the high-SNR capacity expansion, which the authors refer to as phase noise number, is defined as
(10)$$\chi =\lim_{\rho \to \infty }\left(C\left(\rho \right)-0.5\ln\left(\rho \right)\right),$$
where the first term in the right hand side of (Equation 10) is the capacity of phase noise channel and ρ is the SNR. It is concluded that in the uplink the phase noise number is larger when separate oscillators are used. However, the phase noise number of the downlink channel is larger when a common oscillator is used for all the antennas. In [75], the authors have investigated the effects of phase noise on the performance of massive MIMO systems when the number of BS antennas (*M*) and number of users (*K*) are large but their ratio $\beta =\frac{M}{K}$ is fixed. It is assumed that the number of active oscillators is $M_{osc}$ and $\frac{M}{M_{osc}}$ antennas are attached to each oscillator. The analysis show that when β is small, the performance of the system when all the BS antennas are connected to a single oscillator is better than the case where each BS antenna has its own oscillator. However, when β is large and at low SNR, we have the opposite behavior. Table 2 summarizes the effect of the mentioned hardware impairments and their possible solutions.

## 6. Time Varying Channels

With the emergence of high speed vehicles such as high speed trains, one of the targets of 5G is to manage fast mobility communications [76,77,78]. Since massive MIMO is a key technology in 5G, and the performance of massive MIMO system is very dependent on accurate channel knowledge, it is important to study the effect of channel errors and channel variations (and, consequently, channel aging effects) on the performance of massive MIMO schemes, especially in the presence of high speed environments [76]. The studies done on massive MIMO systems, considering the channel dispersion type, can be divided into four categories. In the first category the channel is assumed flat fading and time-invariant. In fact the channel is modeled as independent flat fading whose gain remains unchanged during the coherence time and then changes independently. In the second category the channel is time varying flat fading, where the channel coefficients change symbol-by-symbol due to Doppler effects. In the third category the channel is time-invariant frequency selective. In this case the channel has multi-path effects and its coefficients remain unchanged during a block. The fourth category, which is the most probable one in fast moving environments, corresponds to a time-varying, frequency selective channel, where the channel changes within a block due to Doppler effects. In the following we will focus on the performance of massive MIMO systems in the second and the fourth category.

### 6.1. Time Varying Flat Fading Channels

In this subsection, the channel is modeled as quasi-static block fading where due to Doppler effect the channel coefficients change symbol-by-symbol based on the Jake’s model (the generalization to other Doppler models is more-or-less straightforward). In high speed environments, it is very probable that the channel coefficients change between the time they are estimated and the time they are used by the precoder or equalizer. This phenomenon is called channel aging and degrades the performance of massive MIMO systems. In [79], the sum-rate of a multi-cell massive MIMO system in the presence of channel aging has been analyzed. The analysis were done both in the uplink and the downlink and for both MF and ZF detectors and precoders, respectively. The results indicate that channel aging degrades the performance of the system, as expected. However, with some channel prediction techniques the effect of channel aging can be mitigated. The authors in [80] have considered a multi-cell massive MIMO system with RZF precoder and investigated the effect of channel aging on the performance of the system. In [81], the effects of channel aging on the sum-rate and spectral efficiency of massive MIMO systems with ZF and MF precoders were studied. The results showed that channel aging causes loss in both the sum-rate and the spectral efficiency of the system. Moreover, the impact of channel aging on ZF precoder is much larger than the MF precoder. In [82], the effect of channel aging on the achievable sum-rate of an FDD massive MIMO system was studied, as the dimension of the system increased. MF and MMSE detectors were considered in the uplink and MF and RZF precoders were considered in the downlink analysis. The results indicate that, if the frame duration is properly selected, the effect of channel aging can be substantially mitigated. It was also showed that in high speed environments, using all BS antennas can result in very small achievable rates.

### 6.2. Time Varying Frequency Selective Channels

In this subsection we consider multi-path channels were the channel coefficients change within a block due to Doppler. We name these channels doubly dispersive since the channel has time dispersion due to its multi-path nature and frequency dispersion due to its time variation. In [83], a single-carrier massive MIMO system in doubly dispersive channel has been considered. It is assumed that this system is operating in millimeter wave band and TDD mode. An adaptive channel estimation technique based on reduced rank Kalman filtering has been proposed to cope with channel variation effects. To reduce the channel estimation complexity a pre-beamformer, which considers the channel variation in time, was proposed. In massive MIMO systems, pilots are used for channel estimation. Since the channel changes in time, another pilot block has to be sent again after a while. When the channel time variations are high, the time distance between two consecutive pilot blocks reduces which leads to low spectral efficiency. In [84], the authors investigated the possibility of increasing the time distance between pilot blocks with the aid of channel prediction. Therefore, a channel prediction technique based on Kalman filter and autoregressive moving average (ARMA) channel time variation model was proposed. Simulation analysis shows that for high Doppler frequencies, the use of channel prediction increases the rate compared to the case where we have no channel prediction. However, for low Doppler frequencies, channel prediction improves the performance only when a long time has passed from the last channel estimation. The downlink performance of SC-FDP and OFDM massive MIMO systems in doubly dispersive channels were analyzed in [85]. The sum-rates of both systems were derived analytically in doubly dispersive channels, considering the effect of channel aging and channel estimation error due to Doppler drifts. The analytical and simulation results indicate that SC-FDP is more robust to channel double dispersiveness and channel aging effects than OFDM. In fact, when we have no channel estimation errors due to Doppler drifts, the sum-rate of SC-FDP massive MIMO, unlike OFDM massive MIMO, increases unlimitedly as the number of BS antennas tends to infinity. However, when we have channel estimation errors the sum-rate of both systems saturates when the number of BS antennas gets large with SC-FDP having a slightly higher sum-rate. Figure 9 shows the sum-rate in four scenarios versus the number of BS antennas. In this figure, scenarios I, II, and III refer to the case where we have double dispersive channel with perfect CSI, double dispersive channel with channel aging, double dispersive channel with channel aging, and channel estimation error. In scenario III.A the channel estimation error is modeled with additive noise and in scenario III.B the effect of Doppler shift on channel estimation has been considered. A summary of channel aging and channel time variation effects in massive MIMO systems is presented in Table 2.

## 7. Conclusions

Massive MIMO is one of the key technologies for fifth generation cellular communications. With massive MIMO, for each cell a BS with large number of antennas communicates simultaneously with multiple users using the same physical layer resources. This allows high spectral and power efficiencies. However, this technology faces some challenges which need to be addressed. In this paper we studied the hardware impairment and channel variation and aging effects on both OFDM and SC-FDP waveforms, for massive MIMO systems. This paper gives an overview on the harware impairment and channel aging effects in massive MIMO systems, emphasizing the main works done in this field.

## Figures and Tables

**Figure 1 sensors-19-00164-f001:**
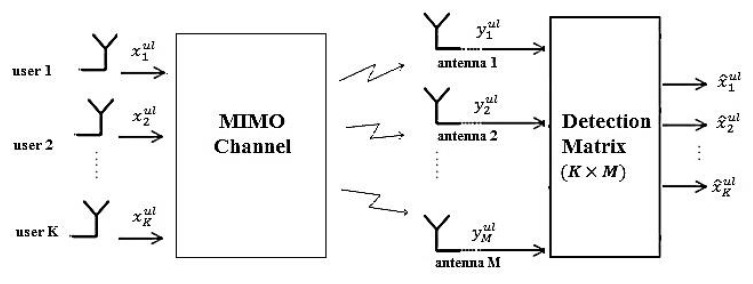
Block diagram of massive multiple input multiple output (MIMO) system in uplnik.

**Figure 2 sensors-19-00164-f002:**
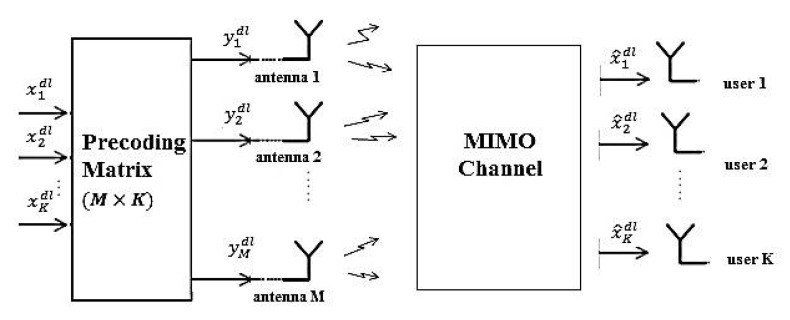
Block diagram of massive MIMO system in downlink.

**Figure 3 sensors-19-00164-f003:**
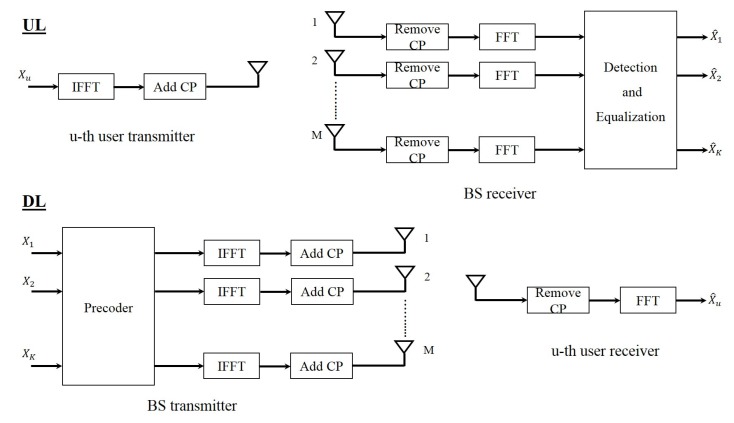
Block diagram of orthogonal frequency division multiplexing (OFDM) massive MIMO system.

**Figure 4 sensors-19-00164-f004:**
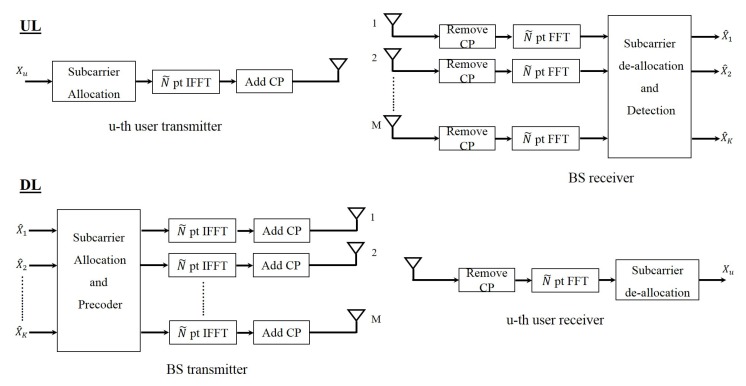
Block diagram of OFDM multiple access (OFDMA) massive MIMO system.

**Figure 5 sensors-19-00164-f005:**
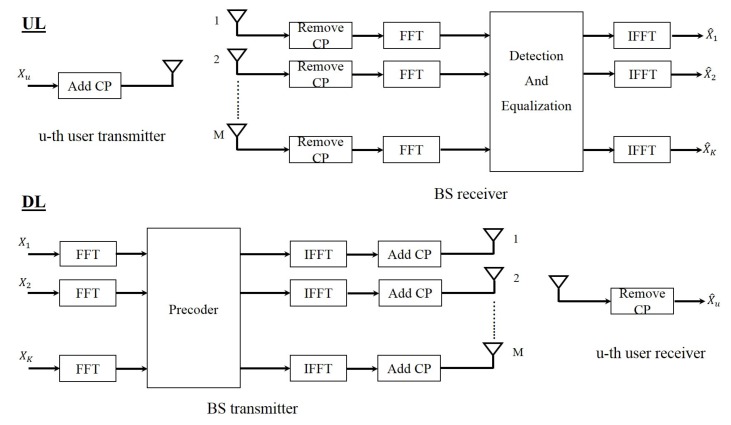
Block diagram of SC-FDP Massive MIMO system.

**Figure 6 sensors-19-00164-f006:**
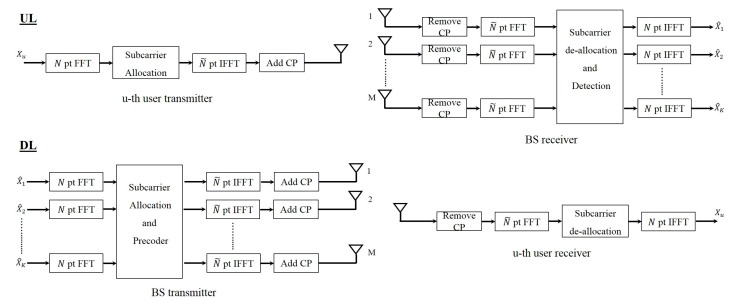
Block diagram of single-carrier frequency domain processing (SC-FDMA) massive MIMO system.

**Figure 7 sensors-19-00164-f007:**
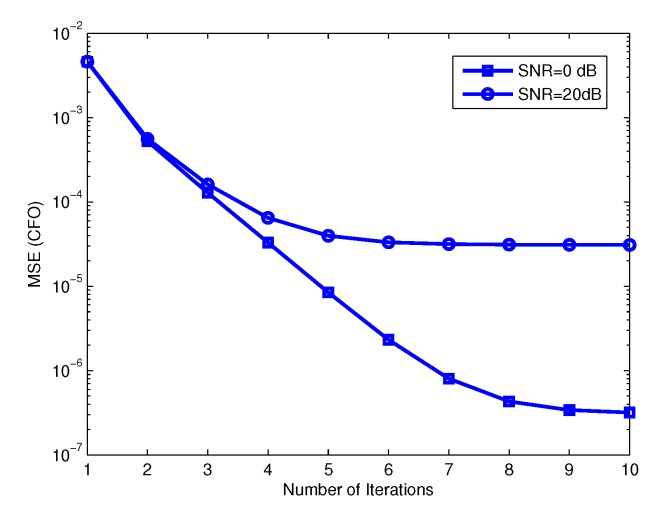
Mean square error (MSE) of normalized carrier frequency offset (CFO) versus number of iterations.

**Figure 8 sensors-19-00164-f008:**
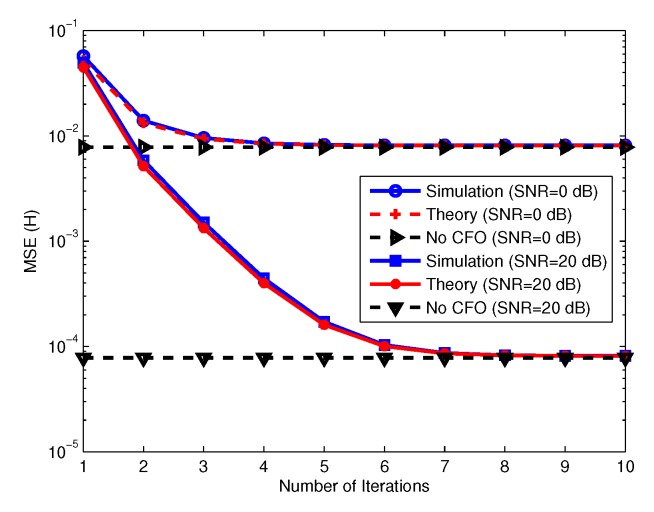
MSE of channel coefficients versus number of iterations.

**Figure 9 sensors-19-00164-f009:**
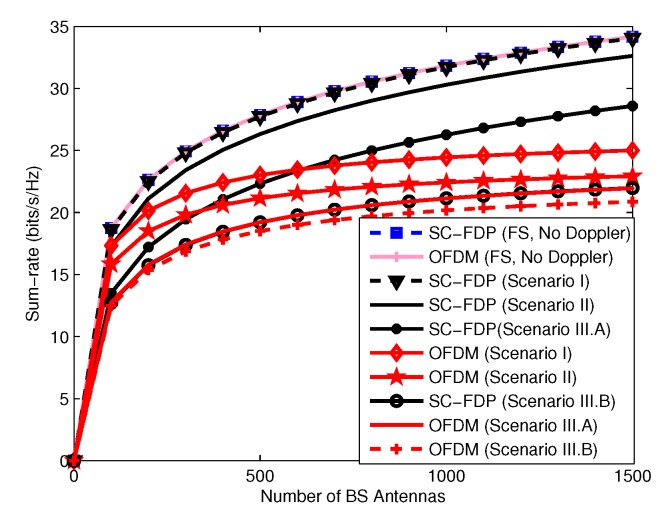
Sum-rate versus the number of BS antennas.

**Table 1 sensors-19-00164-t001:** Surveys on massive multiple input multiple output (MIMO) challenges.

Challenge	Paper
Waveform	[16]
Detection and Precoding	[18,19]
Hardware impairment	This survey
Channel measurement and modeling	[17]
Channel aging and time variation	This survey
Pilot contamination	[20]

**Table 2 sensors-19-00164-t002:** Effect of channel and hardware impairments and their possible solutions.

Impairment	Effect	Possible Solution
CFO	- Performance degradation	- Using low complexity CFO estimation techniques
	- Sum-rate limitation in the OFDM case	- Using less sensitive waveforms such as SC-FDP
PAD	- Performance degradation	- Using Precoders that produce constant envelop signals
	- Sum-rate limitation	- Using less sensitive waveforms such as SC-FDP
	- Out of band radiation	- Using clipping and system degree of freedom to compensate clipping distortion
I/Q Imbalance	- Performance degradation	- Using Precoders that analyze real and imaginary parts separately
	- Sum-rate limitation	
	- Inaccurate channel estimation	
Phase noise	- Performance degradation	- Using more robust precoders
	- Channel aging	- Using phase noise tracking techniques
Channel Aging	- Performance degradation	- Using channel prediction techniques
	- Sum-rate limitation in the OFDM case	- Using less sensitive waveforms such as SC-FDP

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
