# Peer review of "A Survey on Massive MIMO Systems in Presence of Channel and Hardware Impairments"

_sensors, 2019, doi:10.3390/s19010164_

Round 1
Reviewer 1 Report
Overall, this is a nice, interesting and well-written survey on massive MIMO toward real-world implementations. Therefore, I am positive toward its publication within journal. However, before recommending it for publication, it is my opinion that the following aspects should be addressed to improve the paper further:
1) The abstract should end with a brief mention to wrap-up conclusions.
2) “Therefore, massive MIMO systems can have very high spectral and power efficiencies” -> “Therefore, massive MIMO systems can bring very high spectral and power efficiencies.”
3) It would be useful if the authors could add a table which shows the advances provided by the present survey with respect to existing ones.
4) Please add a notation paragraph at the end of Sec. I.
5) The following related works regarding the successful application massive MIMO to WSNs should be considered:
[R1] "Massive MIMO for Decentralized Estimation of a Correlated Source." IEEE Trans. Signal Processing 64.10 (2016): 2499-2512.
[R2] "Massive MIMO channel-aware decision fusion." IEEE Transactions on Signal Processing 63.3 (2015): 604-619.
[R3] "Massive MIMO for Wireless Sensing With a Coherent Multiple Access Channel." IEEE Trans. Signal Processing 63.12 (2015): 3005-3017.
[R4] "Massive MIMO meets decision fusion: Decode-and-fuse vs. decode-then-fuse." Sensor Array and Multichannel Signal Processing Workshop (SAM), 2014 IEEE 8th. IEEE, 2014.
[R5] "Massive MIMO for distributed detection with transceiver impairments." IEEE Transactions on Vehicular Technology 67.1 (2018): 604-617.
6) By reading the whole survey, the section “waveform selection” makes the work a little bit umbalanced with respect to the contributions described in both the abstract and the introduction.
7) For both Secs. 5 and 6, it would be useful adding a table or an illustration which wraps up the effects of each of the considered non-idealities/realistic effects in massive MIMO and a possible solution to cope with them.
Reviewer 2 Report
In this article, authors present a survey on the performance of massive mimo systems under non ideal channel and hardware impairments. In general, the tutorial is well written however I have the following concerns.
a) It is not clear from the title that the paper performs a survey. Therefore the title should be updated to reflect this fact.
b) In my opinion it would be more interesting to somehow present in the current tutorial the available mathematical models for hardware impairments and non ideal channels. It is OK to cite all previous research works but if there are no space constraints, such an addotion would enhance the presentatio on the topic.
Round 2
Reviewer 1 Report
Overall, this is a nice, interesting and well-written survey on massive MIMO toward real-world implementations. Additionally, the authors have satisfactorily addressed all my comments and modified the manuscript in accordance to the reply provided.
Therefore, I am glad to recommend the present paper for publication within this journal.
Reviewer 2 Report
The authors have adressed all my comments